# Improving the Treatment Effect of Carotenoids on Alzheimer’s Disease through Various Nano-Delivery Systems

**DOI:** 10.3390/ijms24087652

**Published:** 2023-04-21

**Authors:** Wenjing Su, Wenhao Xu, Enshuo Liu, Weike Su, Nikolay E. Polyakov

**Affiliations:** 1Collaborative Innovation Center of Yangtze River Delta Region Green Pharmaceuticals, Zhejiang University of Technology, Hangzhou 310014, China; 2Institute of Solid State Chemistry and Mechanochemistry, 630128 Novosibirsk, Russia; 3Institute of Chemical Kinetics and Combustion, 630090 Novosibirsk, Russia

**Keywords:** carotenoids, Alzheimer’s disease, nanosized drug delivery systems

## Abstract

Natural bioactive compounds have recently emerged as a current strategy for Alzheimer’s disease treatment. Carotenoids, including astaxanthin, lycopene, lutein, fucoxanthin, crocin and others are natural pigments and antioxidants, and can be used to treat a variety of diseases, including Alzheimer’s disease. However, carotenoids, as oil-soluble substances with additional unsaturated groups, suffer from low solubility, poor stability and poor bioavailability. Therefore, the preparation of various nano-drug delivery systems from carotenoids is a current measure to achieve efficient application of carotenoids. Different carotenoid delivery systems can improve the solubility, stability, permeability and bioavailability of carotenoids to a certain extent to achieve Alzheimer’s disease efficacy. This review summarizes recent data on different carotenoid nano-drug delivery systems for the treatment of Alzheimer’s disease, including polymer, lipid, inorganic and hybrid nano-drug delivery systems. These drug delivery systems have been shown to have a beneficial therapeutic effect on Alzheimer’s disease to a certain extent.

## 1. Introduction

Alzheimer’s disease (AD) is the most common form of dementia, and has become the main health problem of the elderly [1]. According to a recent report, over 46 million people suffer from Alzheimer’s disease, and that number is expected to grow to 130 million by 2050. AD is a recessive disease, one which generally occurs in people more than 55 years old and worsens with age [2,3]. AD is accompanied by progressive and irreversible dementia, memory loss, pessimism and other behavioral changes, loss of social status, speech loss without changing sensorimotor function, and additional symptoms [4,5]. Although the exact biological facts of Alzheimer’s disease are still unclear, current research shows that the etiology of Alzheimer’s disease is multifactorial; such factors as oxidative stress, loss of synaptic neurons, low levels of acetylcholine, β-Amyloid aggregation and metal accumulation all contribute to the development of Alzheimer’s disease [6]. Amyloid plaques can aggregate into senile plaques on the outer surface of blood vessels and brain neurons, and into intracellular aggregates of neurofibrillary tangles arising from hyperphosphorylated tau [7,8,9,10]. The formation of reactive oxygen species (ROS) is a key factor in the development of AD. ROS can attack and destroy proteins, DNA, lipids and other macromolecules of living cells [11].

Carotenoids are naturally occurring pigments with powerful antioxidant properties [12,13,14], which are secondary metabolites produced by various enzymatic reactions [15,16]. Carotenoids have the ability to quench reactive oxygen species, free radicals and singlet oxygen. Their strong antioxidant power is primarily due to the presence of long-chain conjugated olefins in their structure, which makes them ideal candidates for scavenging free radicals [17,18]. Dietary intake of carotenoids has been reported to be associated with the prevention and treatment of many diseases, such as cardiovascular disease, cancer and age-related macular degeneration [19,20], with particular contributions to the prevention of brain diseases [21]. It plays a crucial role in the human brain and has a variety of roles in animals and plants [22,23,24]. Despite this, bioavailability and stability are major challenges for these natural compounds. Fortunately, structural modifications of these compounds can improve their biological function [25]. One of the approaches that makes it possible to overcome these disadvantages of carotenoids and significantly improve their bioavailability and stability is the use of drug delivery systems [26,27,28,29,30,31]. Recently, the development of nano-drug delivery systems has created exciting opportunities for the prevention and treatment of AD. Heretofore-poorly-distributed drugs are now prepared in nano-drug delivery systems. The drug delivery system has a favorable interaction with endothelial microvascular cells at the blood–brain barrier and can produce elevated concentrations of the drug in the brain parenchyma. These nano-drug delivery systems can achieve better efficacy and safety [32,33,34,35].

## 2. Carotenoids in Alzheimer’s Disease Treatment

### 2.1. Pathogenesis of Alzheimer’s Disease

The etiology of Alzheimer’s disease remains unclear due to the multi-factorial nature of the disease’s process. The formation of amyloid plaques is a key factor in the development of AD. These plaques can aggregate into senile plaques on the outer surface of blood vessels and brain neurons, and in intracellular aggregations of neurofibrillary tangles generated from hyperphosphorylated tau [7,8,9,10,36]. Naturally, oxidative stress (OS) is a major feature of AD. The reasons why neurons are extremely sensitive to OS include the following: (1) the energy generated by oxidative phosphorylation in neuronal mitochondria is extremely important [37,38]; (2) about 20% of the oxygen generated by respiration is used by neurons, 1–2% of which is converted into reactive oxygen species to cause OS [39,40,41,42]; (3) metal ions in neurons accumulate and catalyze ROS production in the brain as the aging process progresses [43]; (4) polyunsaturated fatty acids in neurons are susceptible to oxidation [44]; and (5) neurons have relatively low levels of antioxidants and related enzymes [45,46]. In general, OS level increases with age and is an important factor in inducing AD. ROS has the ability to attack and destroy a variety of macromolecules, including proteins, DNA and lipids of living cells [11,47,48,49,50].

### 2.2. Overview of Carotenoids

Carotenoids (Figure 1) are the most prevalent class of isoprenoid yellow-orange pigments that can be synthesized by photosynthetic organisms and fungal microorganisms and bacteria [51,52]. It can be categorized into two main groups: (1) nonpolar carotenes, such as β-carotene and lycopene, which are the hydrocarbon compound that carry no functional groups [53,54]; (2) polar xanthophylls, for instance, astaxanthin, lutein, and canthaxanthin, the structures of which contain hydrogen, carbon, and oxygen [17]. Carotenoids can also be divided into pro- and non-pro vitamin A, which cannot be converted into retinoids [53]. Carotenoids are known to be efficient compounds due to their antioxidant properties and nontoxic nature, which can minimize the risk of age-related muscular disorders [55].

### 2.3. Therapeutic Mechanisms of Carotenoids on Alzheimer’s Disease

The interest in carotenoids has increased dramatically over the last decade due to their newly discovered activities, in particular, their neuroprotective properties. Neuroprotective mechanisms of carotenoids include antioxidant, anti-inflammatory, and anti-apoptotic activities, as well as the potential to promote neural plasticity. Although the exact molecular mechanisms of neurodegenerative diseases are still being elucidated, aging is considered as a primary risk factor for their development, including development of Alzheimer’s disease [56]. It is usually accepted that increased inflammation and oxidative stress within the brain contribute to neurodegeneration. The brain is usually susceptible to higher oxidative stress due to its high metabolic activity and the presence of various oxidized compounds. Oxidative stress can harm biomolecules (peptides, lipids and so on) and lead to neuronal dysfunction over time. The antioxidant activity of carotenoids is perhaps the known property responsible for its health benefit in prevention of neurodegenerative diseases.

Carotenoids are known to act as singlet oxygen quenchers and free radical scavengers and are used to combat oxidative stress in organisms. Singlet oxygen quenching, important in photosynthesis, relies on the energy transfer between electrophilic singlet oxygen and the carotenoid skeleton. The scavenging rate increases with the conjugation length. There are three main types of carotenoid radical scavenging reactions: (1) electron transfer between free radicals and carotenoids resulting in the formation of carotenoid radical cations or carotenoid radical anions; (2) formation of free radical adducts; and (3) hydrogen atom transfer to form neutral carotenoid group [19]. The structure of carotenoids gives them a strong antioxidant capacity, which protects cells against OS mediated by a variety of stressors by reducing DNA damage and activating the endogenous antioxidant enzymes, including superoxide dismutase and catalase. Promoting the efficacy of endogenous antioxidant enzymes is considered to be an important mechanism of some carotenoids’ action in neuroprotection in the brain [57].

As an example, some authors reported successful application of carotenoid astaxanthin in the reducing of neurotoxicity induced by the amyloid-β fragments in cell culture models of Alzheimer’s disease [58,59,60]. The authors report that astaxanthin protected primary hippocampal neurons from amyloid-β induced ROS generation and calcium dysregulation. Another example of effective anti-AD carotenoid is lycopene; see, for example, review [61]. In vitro studies have shown that pretreatment with this carotenoid reduces amyloid-β induced cellular damage and prevented amyloid-β 1–42-stimulated cellular apoptosis via the inhibition of ROS production, reducing mitochondrial dysfunction and expression of pro-apoptotic factors, and thus inhibiting pro-inflammatory response in microglia. In vivo studies on animal models have replicated the results observed in cell cultures. Humans with AD have significantly lower plasma concentrations of lycopene than do of healthy subjects [62]. Administration of the carotenoid-enriched products in humans is useful in increasing serum carotenoid levels and reducing OS, and thereby preventing neurodegeneration and delaying the onset of dementia.

## 3. Carotenoid-Loaded Nanocarriers for Alzheimer’s Disease Therapy

The brain is a special organ that is protected by two major barriers, the blood–brain barrier (BBB) with its 20 m^2^ surface area, and the blood–cerebrospinal-fluid barrier (BCSFB) [63]. Because of the large molecular weight of carotenoids, they do not penetrate easily to the brain, which presents one of the most important challenges in the development of drugs for the central nervous system [64]. Recently, the development of nano-based drug delivery systems has created exciting opportunities for the prevention and treatment of AD. Heretofore-poorly-distributed drugs are now prepared using nano-drug delivery systems. The drug delivery system has an excellent interaction with endothelial microvascular cells at the blood–brain barrier and is capable of producing elevated drug concentrations in the cerebral parenchyma. First, nanocarriers can pass through the blood–brain barrier passively (through the direct plasma membrane) or actively (endocytosis, pinocytosis, etc.) through transmembrane channels. Secondly, functional groups on the surface of nanocarriers (polysorbate surfactant layer or covalent binding of apolipoprotein, etc.) can enhance the efficiency of the carrier system in penetrating through the blood–brain barrier. The nanocarrier system is easily phagocytized by mononuclear phagocyte system, and then degraded or metabolized by lysosomes. The chemical groups on the surface of inorganic nanomaterials can be metabolized by enzymes or non-enzymes. Organic nanomaterials may first decompose and then metabolize into smaller particles. In the liver, if too large to pass through the pores between the transcellular hepatic sinusoidal endothelial cells, small-particle-sized materials can pass through the pores into the perisinusoidal space and then into the hepatocytes, where they are subsequently passed by monooxygenases, transferases, esters metabolism by enzymes, and epoxide hydrolases [65]. Therefore, these nano-drug delivery systems can achieve better efficacy and safety. Nano-delivery systems such as polymer/biopolymer nano-carriers, lipid-based nano-carriers, inorganic nano-carriers, and hybrid nano-carriers have been used (Figure 2). In the following subsections, a brief overview of nanocarriers will be made.

### 3.1. Polymeric Nanocarriers

Polymer nanocarriers based on biodegradable and biocompatible properties include polymer micelles, polymer nanoparticles, and dendrimers. It is an ideal drug delivery system for carotenoids for the prevention and treatment of AD. Nanoparticles made of arabinogalactan, whey proteins, poly (butyl cyanoacrylate), casein, starch, and so on have been widely studied for drug delivery in the AD [33,35].

#### 3.1.1. Polymeric Micelles

Micelles are core-shell structures formed spontaneously by amphiphilic molecules in water [66]. Proper micelles can be obtained by adjusting the fraction of monomers in the bulk copolymer so that most hydrophobic drugs can be easily incorporated into the core of the micelles. The function of the micellar shell is to protect the drug from interactions with serum proteins and non-target cells. In addition, targeted drug delivery can be achieved by end-functionalization of micelles with block copolymers of peptides, sugars, and additional components. Nanoscale micelles minimize the clearance of micelles from the body, prolong the action time of the drug, and improve the bioavailability of the drug. For instance, micelles prepared from Pluronic block copolymers have been the most studied. Both in vivo and in vitro experiments have shown that the micelle has the ability to enhance the drug’s ability to penetrate the blood–brain barrier [67]. In summary, micelles can respond to external or internal stimuli and thus play a role in stabilizing, targeting or controlling the release of drugs [68,69], which are ideal drug delivery systems for the treatment of central nervous system diseases, particularly AD, due to their excellent ability to penetrate the blood–brain barrier [70,71,72,73].

#### 3.1.2. Polymeric Nanoparticles

Polymer nanoparticles and micelles have some similar characteristics, such as loading efficiency, versatility, stimulus response (including light, temperature, enzyme, pH, and other biological and chemical agents) and so on [74]. Amphiphilic polymers with different structures, lengths and charges can be used to prepare polymer nanoparticles. They vary in size, shape and stability and can be used to encapsulate hydrophilic and hydrophobic drug molecules, including macromolecules such as carotenoids [28,31,75]. Polymer nanoparticles are widely used as biodegradable materials in the medical field. Commonly used polymers include polylactides, polyglycolides, poly-ε-caprolactone, and polyethylene glycol. Although these materials have been approved by the FDA for use in the medical field, they are not considered ideal for the treatment of central nervous system disorders due to their poor solubility and degradation in acidic byproducts. Acrylic polymer nanoparticles, especially poly (butyl cyanoacrylate) (PBCA) nanoparticles, have been widely used in the delivery of drugs in the central nervous system [76]. PBCA can be rapidly degraded in vivo to reduce toxicity due to polymer accumulation in the central nervous system. Drugs used to treat diseases of the central nervous system through the PBCA nanoparticle delivery system include doxorubicin, temozolomide, methotrexate, etc. [77,78]. Compared to acrylic polymers, polyester nanoparticles may be a safer choice for brain drug delivery because the degradation products are mainly water and carbon dioxide [79]. The drug delivery systems of polymer nanoparticles prepared by arabinogalactan [28], polylactic acid and poly (lactide co glycolide) also have been used for drug delivery in the central nervous system [80,81].

#### 3.1.3. Dendrimers

Dendrimers are highly branched molecules with a 3D structure consisting of repetitive monomeric units with highly branched structures [82]. The modifiability of its surface structure gives dendrimers versatility, and the presence of a hydrophobic core enables encapsulation of genes, nucleic acids, and other drug molecules through electrostatic interactions or conjugation for the treatment of central nervous system disorders such as AD [64,83,84]. Polymeric dendrites have been developed for the treatment of Alzheimer’s disease, Parkinson’s disease, multiple sclerosis, ischemic stroke and other central nervous system disorders. Although dendrimers have a well-defined structure, their deeply branched and cross-linked nature makes it difficult to predict their degenerate outcome. Therefore, their inherent toxicity needs to be confirmed by other in vivo studies [85,86,87].

### 3.2. Lipid-Based Nanocarriers

#### 3.2.1. Liposomes

One of the drug delivery systems that has received increasing attention is the liposome, which is a spherical vesicle composed of a unilamellar or multilamellar phospholipid bilayer [88]. Liposomes have excellent biocompatibility and biodegradability, low toxicity, and enable the targeted delivery of lipophilic and hydrophilic drugs. The greatest advantage of liposomes for central nervous system (CNS) delivery is that they can be easily surface-modified to prepare advanced liposomes such as immunoliposomes for targeted delivery [88]. For example, immunoliposomes carrying anti-epidermal growth factor receptor (anti-EGFR) antibodies can enhance the ability of vinorelbine and doxorubicin to target brain tumor cells [89]. Although liposomes have the problems of rapid clearance rate in vivo, low stability and inability to achieve long-term drug release, these problems have been solved step by step. For instance, modification of liposomes with polyethylene glycol (PEG) can reduce the possibility of liposomes being engulfed by macrophages and prolong their time in blood circulation, and the surface-functionalization of liposomes can improve their stability and efficacy. It is expected that additional liposomes will be used for the delivery of drugs in the central nervous system [90,91].

#### 3.2.2. Solid Lipid Nanoparticles (SLNs)

SLN is normally composed of a lipid matrix in the solid state at room temperature and dispersed in water or a solution composed of surfactants for stabilization at body temperature. Fatty acids, cholesterol, monostearin, etc. are commonly used lipid matrices for the preparation of SLNs [92,93]. Solid lipid nanoparticles are produced from natural materials or natural lipids, are biocompatible, and do not affect the internal and external environment of cells after degradation, which makes them less immunogenic. At the same time, solid lipid nanoparticles are modest in size, flow rapidly in the blood, and are not easily absorbed by macrophages, thus facilitating the continuous release of therapeutic drugs in the body. Because solid lipid nanoparticle delivery systems are able to bypass P-glycoprotein by cell-by-cell percolation, they facilitate the penetration of lipophilic drugs across the blood–brain barrier. In fact, SLN can also bind apolipoprotein to target brain tissues [94,95]. Bhatt et al. prepared astaxanthin solid lipid nanoparticles by a double lotion solvent displacement method and administered the drug through the nose to increase the efficiency of brain targeting. The results of biological distribution experiments indicate relatively high concentrations of the drug in the brain and that intranasal administration of astaxanthin maximizes nerve protection against oxidative stress [96]. Therefore, solid lipid nanoparticles have become the best candidate drug delivery system for CNS drugs. Their low encapsulation efficiency due to the lipid core, which does not allow for the generation of empty space to encapsulate the underlying material in the solid phase again during crystallization, limits the large-scale production and development of solid-state lipid nanoparticles [70,97,98,99].

#### 3.2.3. Nanostructured Lipid Carriers (NLCs)

To improve the inherent shortcomings of SLNs for high load efficiency, nanostructured lipid carriers came into being [100]. Nanostructured lipid carriers are composed of liquid and solid lipids (inner layer) and water emulsifiers (outer layer). They are transformed forms of SLN. The difference between SLN and NLC is that in NLC, 5–40% of the solid phase is exchanged with the liquid phase [101], liquid lipids or oils including fatty acid esters or alcohols such as 2-octanol are mixed with solid lipids [102,103], and lipids with different chain lengths, such as mono-, di-, and triglycerides, are also used to increase the space of the delivery system [104]; note that some hydrophobic drugs have better solubility in liquid lipids, which makes nanostructured lipid carriers more loading-efficient [105,106]. Rodriguez Ruiz et al. used sunflower seed oil as liquid lipid to synthesize astaxanthin loaded nanostructured lipid carriers by a green method. The results showed that nanostructured lipid carriers could stabilize astaxanthin molecules and maintain and enhance its antioxidant activity [107].

### 3.3. Inorganic Nanocarriers

Inorganic nanocarriers include cerium dioxide, iron oxide, gold, inorganic quantum dots and so on. Inorganic nano-delivery systems have been used for drug delivery due to their excellent physical and chemical properties, including size, shape, surface functionality, chemical structure, and high specific surface area, as well as imaging capabilities, so they can play their therapeutic and diagnostic roles at the same time. However, their biocompatibility, biodegradability and safety become the difficulties that restrict their wide application [108,109,110].

### 3.4. Hybrid Nanocarriers

Hybrid nanocarriers are composed of lipid, organic, and inorganic polymers [97,111,112]. In general, the organic polymer or inorganic substance acts as the core, and the lipid layer acts as the shell. The outer lipid can inhibit the diffusion of water to the inner layer, thus delaying the degradation of the inner polymer and ensuring that the loaded drug molecules are slowly and continuously released. Hybrid nanocarriers have controllable drug release capacity, high loading efficiency, biocompatibility and biodegradability, and are good carriers for the treatment of central nervous system diseases [97]. For instance, gold nanoparticles have been coated with PEG through pH-sensitive hydrazine bonds and modified with gadolinium-chelate and lipoprotein receptor-related protein-1 (LRP-1) recognition peptides. After BBB penetration and PEG cleavage, the mixed nanoparticles aggregated in the acidic tumor environment, resulting in an increased magnetic resonance imaging (MRI) signal [113].

## 4. Different Nano-Encapsulated Carotenoids in Alzheimer’s Disease Therapy

### 4.1. Crocin and Crocetin

Crocin (Figure 1) and crocetin are mainly derived from saffron and the fruits of gardenia. Crocin, a water-soluble carotenoid, is composed of a conjugated polyene skeleton and sugar substituents at both ends. When the substituents at both ends are hydrogen atoms, it is a crocetin, which is insoluble in water. Numerous studies have shown that crocin and crocetin have disparate pharmacological effects, such as anti-Alzheimer, antioxidant, anti-tumor, anti-inflammatory, memory enhancing, antidepressant, etc. [114,115].

Sonali [116] and his colleagues obtained Crocus sativus extract from Crocus sativus (stigma) by using the cold-maceration method. The main ingredient in the extract is crocin. They combined extract with various excipients (hydroxypropyl methyl cellulose, ethyl cellulose, and Eudragit), which were prepared by physical mixing into particles, and then filled into hard gelatin capsules. The results showed that the capsule formulation had better dissolution properties and the in vitro pharmacokinetics showed more plasma exposure volume. The plasma concentration of crocin was increased by a factor of 3.3 compared to the extract. Their study has shown that crocin enhances the clearance of the amyloid-β in an AD brain by inducing the expression of P-gp. Hence, it has therapeutic and protective effects against Alzheimer’s disease.

Another study of crocin application against AD has recently been published by Song et al. [33]. The authors reported that they had successfully made crocin-loaded hollow dextran sulfate/chitosan-coated zein nanoparticles using a layer-by-layer self-assembly technique. Compared with uncoated crocin-loaded zein nanoparticles, coated nanoparticles showed better controlled release effects in vitro (simulated gastrointestinal digestion) and stronger antioxidant activity. Experimental results in AD cell models showed that these nanoparticles can reduce the amyloid-β concentration in differentiated SH-SY5Y cells by 43%. These results prove that dextran sulfate/chitosan-coated crocin-loaded zein nanoparticles could be a hopeful delivery system for treating AD.

To overcome the poor water solubility and bioavailability of crocetin, Wong et al. [117] prepared an innovative water-soluble crocetin-γ-cyclodextrin inclusion complex for intravenous injection. The inclusion complex was shown to be safe in cell experiments. Pharmacokinetic and biodistribution experiments have shown that the inclusion complex can improve the bioavailability of crocetin and achieve the effect of facilitating the crocetin delivery system through the blood–brain barrier to the brain. The pharmacological mechanism of crocetin is through reducing the production of amyloid-β protein.

### 4.2. Astaxanthin

Astaxanthin (AST, 3,3′-dihydroxy-β, β-carotene-4,4′-dione, Figure 1) is a xanthophyll-type carotenoid [118]. It is synthesized by algae, bacteria or yeasts and can also accumulate in birds, fish and crustaceans through the food chain [119]. Unlike other carotenoids, astaxanthin contains two ionone rings at each end of the carbon chain with long-chain conjugated double bonds. Their unique molecular configuration and size allow astaxanthin molecules to be inserted vertically into the phospholipid bilayer of the cell membrane, allowing astaxanthin to prevent lipid peroxidation and protect the integrity of the cell membrane [120]. It is by far the most powerful natural antioxidant found in nature. Its strong antioxidant effect is due to the effective stability of the ionone rings and polyene skeletons against free radicals. The mechanism of antioxidant activity in AST involves the absorption of free radicals into the polyene chain, providing electrons or forming chemical bonds with the active material [121]. In addition to this, astaxanthin has anti-inflammatory, anti-diabetic and anti-cancer activity and cardiovascular disease prevention properties [122], as well as efficacy against nervous system diseases [123]. Because of astaxanthin’s beneficial effects on biological systems, the FDA has approved the use of the pigment as a food colorant in animal and fish feed [124]. It has been assumed that the anti-cancer activity of AST might be related to its ability to chelate metal ions and produce neutral radicals [125,126,127].

Debora et al. formulated stealth lipid nanoparticles loaded with AST as a potential strategy for evading the defense lines represented by macrophages and improving the stability of the drug for the achievement of good bioavailability in the brain [34]. Prakash et al. reported the preparation of astaxanthin solid lipid nanoparticles by solvent displacement using citric acid, lecithin and poloxamer. Astaxanthin solid lipid nanoparticles have shown a strong neuroprotective effect against oxidative stress in neuronal cell lines. Published γ scintigraphy and radiological data showed that 99mTc AST solid lipid nanoparticles were directly transported from nose to brain. These findings confirm that these nanoparticles can be effectively used for brain targeting and can provide protection against a variety of neurological diseases, firmly suggesting that astaxanthin can provide neuroprotection against oxidative stress-induced cell damage [96].

### 4.3. Lycopene

Lycopene, a natural carotenoid, is widely found in fruits as diverse as pink guavas, tomatoes and red-skinned watermelons [128,129]. In the clinic, lycopene is the subject of numerous studies investigating its anti-cancer, cardiovascular disease prevention, liver protection, and other alternative effects. In 2003, Zhao et al. [130] used mice as test subjects and compared lycopene to aspirin, which they found to have similar anti-inflammatory activity. It is precisely because lycopene has neuroprotective activities, such as antioxidant and anti-inflammatory properties, that it is expected to be used to prevent Alzheimer’s disease [131]. However, lycopene has a poor bioavailability and is easily affected by the presence of cis-isomers and other carotenoids [132]. Therefore, an appropriate administrative approach is required. Nano-delivery systems have great potential and advantages in improving bioavailability and drug stability [61].

### 4.4. Lutein

Lutein, a dietary carotenoid, comes from foods such as egg yolk, corn, kiwi, persimmon, and green vegetables. Its structure is conjugated polyene skeleton and ketone at both ends. Due to the particularity of its structure, it is easily affected by environmental factors such as temperature, light, oxygen and so on [133]. Similar to other carotenoids, it has antioxidant, anti-inflammatory and anti-tumor activity, and the potential to protect against several diseases, including Alzheimer’s disease and age-related macular degeneration. However, its poor absorption and bioavailability due to its low solubility limits its use in the food and pharmaceutical industries [134]. Therefore, it is necessary to develop appropriate dosage forms in order to successfully deliver it.

Through electrostatic interaction and nanoprecipitation method, Dhas and Mehta [35] prepared spherical nanoparticles with chitosan as shell and PLGA as core with the particle size less than 150 nm. The nanoparticles were administered through the nasal cavity. This is a relatively short route of drug delivery, allowing the target drug to reach the brain more quickly. In vitro results from the co-culture BBB model show that the nanoparticles are able to penetrate the blood–brain barrier more effectively than pure lutein suspension and PLGA nanoparticles. In addition, in vitro release experiments, in vitro toxicity experiments, and cell uptake experiments have demonstrated, respectively, the effect of sustained release, the safety of nanoparticles, and the ability of vesicle-mediated endocytosis to achieve more efficient delivery. Antioxidant experiments have demonstrated the excellent ROS scavenging activity of nanoparticles. The delivery of lutein from the nasal cavity to the brain can be effective in reducing oxidative stress and thus achieving efficacy in treating AD.

### 4.5. Fucoxanthin

Fucoxanthin, an orange pigment and one of the most abundant carotenoids in nature [135], is found in the chloroplasts of brown algae [136]. Recently, fucoxanthin has been reported to have a variety of biological activities, including anti-cancer, antioxidant, anti-angiogenesis, anti-diabetes, anti-obesity, anti-inflammatory, and anti-malaria activities [137]. Fucoxanthin has previously been reported to have neuroprotective effects against Alzheimer’s disease [32,138,139,140,141]. Fucoxanthin has also been shown to be safe in preclinical and small-population clinical studies, but its low bioavailability in the central nervous system limits its clinical use. To overcome this problem, nanoparticles with a diameter of about 200 nm and negative charge have been synthesized, and their penetration into the central nervous system is recommended [32]. Fucoxanthin nanoparticles can continuously release fucoxanthin in physiological environments. Fucoxanthin nanoparticles showed significant inhibition of amyloid-β formation of fibrils and oligomers. Most importantly, intravenous injection of fucoxanthin nanoparticles can prevent amyloid-β oligomer induced cognitive impairment in AD mice more effectively than fucoxanthin. These results indicate that nanoparticles can improve the bioavailability of drug delivery and enhance its efficacy in the treatment of AD, which may make it possible to use them in the treatment of AD in the future [32].

## 5. Conclusions and Perspectives

Nowadays, it is clear that carotenoids have many benefits for health and positive nutritional effects and can reduce the risk of many diseases. However, there are some critical points to be considered: (1) Most carotenoids play a synergistic role when combined with other compounds, and the single form of carotenoids may not be effective, but from another perspective if two or more carotenoids are put together inside the nanocarrier there may be competition for absorption, which leads to lower bioavailability. (2) Carotenoids are unstable, and easy to transform into different compounds; therefore, the safety of carotenoids needs additional research. (3) The therapeutic effect varies from person to person, thus the effective dose is an unknown problem.

Carotenoid-based nano-drug delivery systems are feasible for effective disease prevention and treatment. This review presents a series of examples of carotenoid nano-delivery systems for Alzheimer’s disease. Each technology has its own strengths and limitations. To some extent, nano-delivery systems can improve the loading capacity, bioavailability, bioactivity, stability and solubility of carotenoids. In the author’s opinion, polymeric micelles are more suitable for the delivery of carotenoids. First, the polymer micelles can be adjusted to a suitable size to accommodate carotenoids of different sizes. Second, the modifiability and ease of modification of the polymer surface increases the functional properties of carotenoids. Finally, carotenoid polymers can make it easier to pass through the blood–brain barrier by adjusting the hydrophilic–lipophilic balance. However, industrial production of nanomedicines is still in its early stages. Safety and health concerns need to be explored in depth before widespread consumption. First, each process or material must be formally approved by regulatory authorities. However, the regulatory framework for the inclusion of nano-carriers in pharmaceutical products is still in flux. State agencies are expected to add initiatives and some legislation to regulate and monitor the proper development and application of nanoparticles in food and drug formulations.

In modern medicine, the idea of “synergy” between drug and carrier has attracted increased attention, seeking to preserve and improve the health benefits of various drugs in prevention and the treatment of many diseases. In the case of carotenoids, drug delivery systems can assist these bioactive compounds in exerting greater biological activity and stability. On the other hand, some nano-delivery systems can also play more functional roles, including targeted delivery to the brain or other organs, or overcome the blood–brain-barrier. Thirdly, since most of the nanocarriers are natural protein or polysaccharide components, they can provide the body with some needed nutrients to a certain extent and improve the efficiency of disease prevention and treatment.

## Figures and Tables

**Figure 1 ijms-24-07652-f001:**
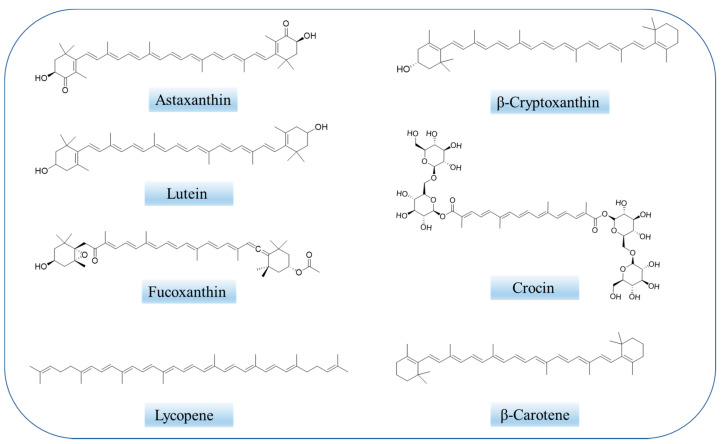
Chemical structures of some carotenoids.

**Figure 2 ijms-24-07652-f002:**
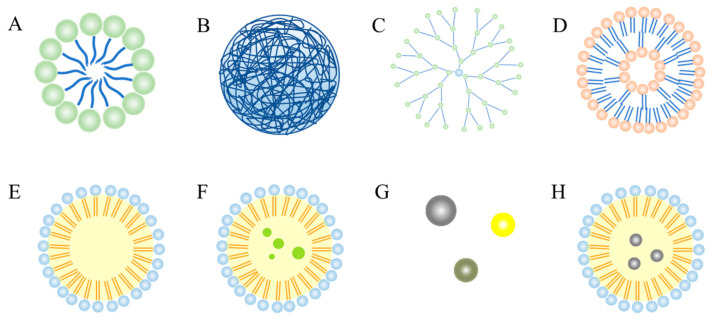
Different types of carotenoids nano-delivery systems for Alzheimer’s disease treatment ((**A**) Polymeric micelles; (**B**) Polymeric nanoparticles; (**C**) Dendrimers; (**D**) Liposomes; (**E**) Solid lipid nanoparticles; (**F**) Nanostructured lipid carriers; (**G**) Inorganic nanocarriers; and (**H**) Hybrid nanocarriers).

## Data Availability

Not applicable.

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
