# Peer review of "Improving the Treatment Effect of Carotenoids on Alzheimer’s Disease through Various Nano-Delivery Systems"

_ijms, 2023, doi:10.3390/ijms24087652_

Round 1
Reviewer 1 Report
In this work improving the treatment effect of carotenoids on Alzheimer's disease through various nano-delivery systems is described. This review contains recent data on different carotenoid nano-drug delivery systems for the treatment of Alzheimer's disease, including polymer, lipid, inorganic and hybrid nano-drug delivery systems. The work is of interest because it was shown that these drug delivery systems have a beneficial therapeutic effect on Alzheimer's disease to a certain extent. Taking into account the mentioned below notes, I think that the article looks like a Review article and may be published after minor revision.
Notes:
1. There are some printing mistakes that should be checked and corrected. For example, lines 131, 174, 194, 213, 227, “in vivo” should be written by italic.
2. Are examples known in the literature on the preparation of drug delivery systems for carotenoids based on dendrimers? Which core, generation and functional groups are best suited for these purposes? Some examples should be presented.
3. Figure 2 should be complemented by the schematic example of hybrid nanocarrier.
4. This review presents a series of examples of carotenoid nano-delivery systems for Alzheimer's disease. Which kind of these systems is more suitable for carotenoid delivery? I think short summarizing about it should be added in conclusion.
Author Response
Reviewer #1:
General comment:
In this work improving the treatment effect of carotenoids on Alzheimer's disease through various nano-delivery systems is described. This review contains recent data on different carotenoid nano-drug delivery systems for the treatment of Alzheimer's disease, including polymer, lipid, inorganic and hybrid nano-drug delivery systems. The work is of interest because it was shown that these drug delivery systems have a beneficial therapeutic effect on Alzheimer's disease to a certain extent. Taking into account the mentioned below notes, I think that the article looks like a Review article and may be published after minor revision.
Answer: The comments are all valuable and highly helpful in revising and improving our paper.
Question 1: There are some printing mistakes that should be checked and corrected. For example, lines 131, 174, 194, 213, 227, “in vivo” should be written by italic.
Answer: Thank you for your careful guidance. We have added the correct writing to the manuscript.
Question 2: Are examples known in the literature on the preparation of drug delivery systems for carotenoids based on dendrimers? Which core, generation and functional groups are best suited for these purposes? Some examples should be presented.
Answer: Thank you for your valuable suggestions. We apologize for the omission of the information. Although dendrimers are widely used drug delivery systems, we could not find an examples of their use for carotenoid delivery to treat Alzheimer's disease.
Question 3: Figure 2 should be complemented by the schematic example of hybrid nanocarrier.
Answer: Thank you for your careful guidance. We are sorry for missing this essential information. We have completed the details in the manuscript.
Question 4: This review presents a series of examples of carotenoid nano-delivery systems for Alzheimer's disease. Which kind of these systems is more suitable for carotenoid delivery? I think short summarizing about it should be added in conclusion.
Answer: Thank you for your valuable suggestions. I'm sorry that we have ignored this essential information. We have completed the details in the manuscript: " In the author's opinion, polymeric micelles are more suitable for the delivery of carotenoids. First, the polymer micelles can be adjusted to a suitable size to accommodate carotenoids of different sizes. Second, the modifiability and ease of modification of the polymer surface increases the functional properties of carotenoids. Finally, carotenoid polymers can make it easier to pass through the blood-brain barrier by adjusting the hydrophilic-lipophilic balance.".
Reviewer 2 Report
The review deals with "Improving the treatment effect of carotenoids on Alzheimer's disease through various nano-delivery systems" The author lists the different carotenoids and nano-delivery systems used for the treatment of Alzheimer's disease. I have a few comments for improving the clarity of the review.
1. In the abstract, the carotenoids listed as be main components of plant pigments are not to be mandatory major plant pigments, kindly check it.
2. how do these nanocarriers pass through the blood-brain barrier and how body metabolize these nanocarriers?
3. The author mentioned about synergistic effects of carotenoids but if 2 or more carotenoids are put together inside the nanocarrier there may be competition for absorption and it leads to lower bioavailability. The line " easy to metabolize also need to be changed as we still don't have a clear picture of individual carotenoid metabolism. kindly reframe the sentence in conclusion 419 - 421.
4. the author mentioned about synergetic effect of nanocarrier and carotenoids but author failed to explain the nutraceutical effect of nanomaterials/nanocarrier ? kindly explain about it in seperate subsection.
Author Response
Reviewer #2:
General comment:
The review deals with "Improving the treatment effect of carotenoids on Alzheimer's disease through various nano-delivery systems" The author lists the different carotenoids and nano-delivery systems used for the treatment of Alzheimer's disease. I have a few comments for improving the clarity of the review.
Answer: The comments are all valuable and highly helpful in revising and improving our paper.
Question 1: In the abstract, the carotenoids listed as be main components of plant pigments are not to be mandatory major plant pigments, kindly check it.
Answer: Thank you for your careful guidance. We have added the correct writing to the manuscript.
Question 2: how do these nanocarriers pass through the blood-brain barrier and how body metabolize these nanocarriers?
Answer: Thank you for your careful suggestions. I'm sorry that we have ignored this essential information. We have completed the details in the manuscript: "First, nanocarriers can penetrate the blood-brain barrier passively (through the direct plasma membrane) or actively (endocytosis, pinocytosis, etc.) through transmembrane channels. Secondly, functional groups on the surface of nanocarriers (polysorbate surfactant layer or covalent binding of apolipoprotein, etc.) can enhance the efficiency of the carrier system to penetrate the blood-brain barrier. The nanocarrier system is easily phagocytized by mononuclear phagocyte system, and then degraded or metabolized by lysosomes. The chemical groups on the surface of inorganic nanomaterials can be metabolized by enzymes or non-enzymes. Organic nanomaterials may first decompose and then metabolize into smaller particles. In the liver, if too large to pass through the pores between the transcellular hepatic sinusoidal endothelial cells, small particle size mate-rials can pass through the pores into the perisinusoidal space and then into the hepatocytes, where they are subsequently passed by monooxygenases, transferases, esters metabolism by enzymes and epoxide hydrolases [65]. Therefore, these nano-drug delivery systems can achieve better efficacy and safety. ".
Question 3: The author mentioned about synergistic effects of carotenoids but if 2 or more carotenoids are put together inside the nanocarrier there may be competition for absorption and it leads to lower bioavailability. The line " easy to metabolize also need to be changed as we still don't have a clear picture of individual carotenoid metabolism. kindly reframe the sentence in conclusion 419 - 421.
Answer: Thank you for your professional and scientific guidance. We have added the correct writing to the manuscript: "However, there are some critical points to be considered: 1) most of carotenoids play a synergistic role when combined with other compounds, and the single form of carotenoids may not be effective, but from another perspective if two or more carotenoids are put together inside the nanocarrier there may be competition for absorption and it leads to lower bioavailability; 2) carotenoids are unstable, easy to transform into different compounds, therefore the safety of carotenoids needs additional research; 3) the therapeutic effect varies from person to person, thus the effective dose is an unknown problem.".
Question 4: the author mentioned about synergetic effect of nanocarrier and carotenoids but author failed to explain the nutraceutical effect of nanomaterials/nanocarrier? kindly explain about it in separate subsection.
Answer: Thank you for your professional and scientific guidance. We have completed the details in the manuscript: "Thirdly, since most of the nanocarriers are natural protein or polysaccharide components, they can provide the body with some needed nutrients to a certain extent and improve the efficiency of disease prevention and treatment.".
Round 2
Reviewer 2 Report
The authors carefully address the reviewers comments and the manuscript can be accepted.